# Uncertainty Analysis and Data Fusion of Multi-Source Land Evapotranspiration Products Based on the TCH Method

Zilong Cui [1,2], Yuan Zhang [1,*], Anzhi Wang [1], Jiabing Wu [1] and Chunbo Li [1]

1    Key Laboratory of Forest Ecology and Management, Institute of Applied Ecology, Chinese Academy of Sciences, Shenyang 110016, China; cuizilong23@mails.ucas.ac.cn (Z.C.); waz@iae.ac.cn (A.W.); wujb@iae.ac.cn (J.W.); lichunbo20@mails.ucas.ac.cn (C.L.)
2    University of Chinese Academy of Sciences, Beijing 100049, China
*    Correspondence: zhangyuan@iae.ac.cn

**Abstract:** Evapotranspiration (ET) is a very important variable in the global water cycle, carbon cycle, and energy cycle. However, there are still some uncertainties in existing ET products. Therefore, this paper evaluates the uncertainty of three widely used global ET products (ERA5-Land, GLDAS-Noah, and MERRA-2) based on the three-cornered hat (TCH) method, and generates a new ET product based on this. The new product is a long-series global monthly ET dataset with a spatial resolution of $0.25° \times 0.25°$ and a time span of 21 years. The results show that ERA5-Land (8.46 mm/month) has the lowest uncertainty among the three ET products, followed by GLDAS-Noah (8.81 mm/month) and MERRA-2 (11.78 mm/month). The new product (TCH) captures ET trends in different regions as well as validating against in situ flux observations, and it exhibits better performance than the re-analysis dataset (ERA5-Land) in vegetation classifications such as evergreen needle-leaf forest, grassland, open shrubland, savanna, and woody savanna. The linear trend analysis of the new product shows a significant decreasing trend in south-eastern South America and southwestern parts of Africa, and an increasing trend in almost all other regions, including eastern North America, north-eastern South America, western Europe, north-central Africa, southern Asia, and south-eastern Oceania.

**Keywords:** ET; TCH; uncertainty analysis; data fusion

## 1. Introduction

Land evapotranspiration (ET), which consists of plant transpiration, soil evaporation, and canopy-interception evaporation, plays a very important role in the global water cycle and energy exchange. ET is the largest global water flux after precipitation, and about 60% of land precipitation is returned to the atmosphere in the form of ET [1]. At the same time, the ET process also consumes about three-fifths of the net radiation energy of the surface [2]. In addition, due to the coupling of carbon and water, accurate ET values also have a greater impact on ecosystem productivity and accurate assessment of agricultural irrigation needs [3]. Moreover, as the global temperature rises, the process of water vapor exchange between the atmosphere, land, and vegetation becomes more and more intense. Therefore, accurate evaluation of ET values and ET trends in different spatial and temporal scales is important for understanding and analyzing regional and global hydrological cycles, extreme events (such as extreme droughts and floods), and climate change [4,5].

Due to the diverse vegetation classifications on the land surface, its complex land–atmosphere–vegetation-water–heat–exchange processes generate large uncertainties, making it more difficult to accurately assess land ET than that of the oceans [6]. At the same time, due to the uneven distribution of observation stations, the small observation range, and the short observation time series, it is difficult to achieve large-scale and long-term land-ET measurements. Therefore, ET products based on machine learning, land-surface models, re-analysis, and remote-sensing inversion algorithms are widely used, which can produce global datasets in different time and spatial scales [7,8]. Specifically, the essence of

most machine learning algorithms is based on ET observation data from flux towers to be elevated to a regional or global scale. The advantage of this algorithm is that the estimation of ET has high precision. However, its high accuracy is guaranteed by a sufficient number of inputs to locate ET observations [9,10]. Moreover, another significant disadvantage of this kind of algorithm is the lack of contribution from physical processes, due to its black-box characteristics [11,12]. In contrast, land-surface models have explicit physical processes that can explain various thermodynamic and aerodynamic mechanisms. However, the land-surface model focuses on the overall correlation between water and energy, and it requires a large number of input variables and parameters unrelated to ET, which leads to large uncertainties in the estimation of ET [13,14]. The re-analysis technique is expected to obtain more accurate ET estimations in different spatiotemporal scales. This technique can be used to combine the model-simulated ET with multi-source observation data in a specific way, which can reduce the estimation error and uncertainty [15,16]. The main disadvantage of the re-analysis technique is that it relies heavily on the numerical simulation of a large amount of atmospheric forcing, and its calculation is difficult [17]. Satellite remote-sensing inversion algorithms can realize ET estimations with high spatial and temporal resolution within short times and with low economic costs. This is considered to be the only viable method for estimating global land ET [18]. However, it has to be pointed out that remote-sensing inversion cannot directly estimate ET, and it is also necessary to invert relevant variables and obtain ET indirectly by combining certain parameter constraints, which leads to the introduction of uncertainty into this process.

Based on the above methods, a number of ET products have been developed and released worldwide in the past few decades. However, the estimates of these existing ET products vary widely, and some products even have the opposite long-term trends in the same regions [19]. Therefore, these ET products need to be compared with each other or with in situ observations to conduct uncertainty analyses, and they can be fused according to the uncertainty relationship between products to form products with less uncertainty. In previous studies, it was found that the differences between these ET products did not hinder the use of these land-ET products, and the differences between them could promote the research of the best fusion method to obtain products with lower uncertainty [20]. However, due to the differences in various algorithms and calibration coefficients, the ET simulation results are bound to be quite different. Through the fusion of various algorithm datasets, relatively high-precision land ET can be obtained. Although this may not explain the algorithm of the new dataset or its ET physical processes, uncertainty can be reduced through the fusion of multiple algorithmic products and more sophisticated data-fusion methods [21–23]. Various fusion techniques, among them maximizing R [24], least-squares merging [25,26], simple Taylor skill score [27], reliability ensemble averaging [6], and Bayesian model averaging [23], have been widely used to fuse datasets to improve their accuracy. However, these fusion methods require the selection of a reference dataset from the input dataset, which may lead to the inherent uncertainty of the original reference dataset after fusion, and previous studies have mostly focused on the fusion and evaluation of ET simulations at regional scales [28]. At the same time, due to the complex structure and high difficulty of many fusion methods, the efficiency of weight calculation is affected, and the scope of application is limited. In contrast, the three-cornered hat (TCH) method, which does not use any reference datasets, is more advantageous in the fusion of multi-source datasets [29]. The TCH method can evaluate the uncertainty of grid scale of different ET products. The fusion product absorbs the advantages of various ET products, and its performance is more balanced, which can be applied to different climatic regions or vegetation types.

The TCH method originated from the original three-cornered (TC) method. The TC method assumes that any two sets of three datasets are independent of each other, that is, the variables are not related to each other. At this point, the difference between the three simulated time series and the variance corresponding to that difference can be estimated. A unique solution can be obtained by establishing the relationship between the difference

in the known simulated time series and the variance of the unknown individual time series. However, when there are more than three simulated time series, the variance of the known time series is greater than the covariance of the unknown simulated time series, and negative variance may occur based on the assumption that the simulated time series are independent of each other [30]. The TCH method takes into account cross-correlation between datasets, does not require datasets to be independent of each other, and can handle related data with appropriate constraints (minimizing global correlation) to solve the above problems [29]. This study aimed to (1) quantify the uncertainty between three ET products using the TCH method; (2) use the uncertainty relationship as a fusion weight to develop a long-term high-quality global land-ET product; and (3) verify the accuracy of the fusion product through the sites' flux data. We evaluated multi-source ET products and generated a set of global ET products with a long time series. The fusion product (TCH) combines the advantages of each original product, effectively reduces the uncertainty of land ET, and can be applied to a variety of regions. Evaluated products and the new product can provide essential basic data for research in the field of land–atmosphere–vegetation interactions. The new product can also be used to study land-ET trends with climate change.

## 2. Data and Methods

In this paper, three commonly used global-scale datasets were selected as input data for the uncertainty analysis and data fusion (Table 1). All data were added up to the monthly scale, and the spatial resolution was 0.25° after resampling. The EC latent heat flux data in the FLUXNET2015 dataset was converted to ET as the validation data of the fusion product.

**Table 1.** Summary of ET datasets for uncertainty analysis and fusion.

| Dataset | Spatial Resolution (°) | Temporal Resolution | Time Span | Citation |
|---|---|---|---|---|
| ERA5-Land | $0.1 \times 0.1$ | 1 h | 2002–2022 | [31] |
| GLDAS2.1 | $0.25 \times 0.25$ | 3 h | 2002–2022 | [32] |
| MERRA-2 | $0.625 \times 0.5$ | 1 h | 2002–2022 | [33] |

### 2.1. Data

#### 2.1.1. The Land Section of the Fifth-Generation ECMWF Re-Analysis (ERA5-Land) ET

ERA5-Land was obtained by reanalyzing the land-surface portion of the ECMWF ERA5 climate re-analysis. Compared to ERA5 and ERA-Interim, it has a higher spatial resolution (0.1°), allowing for more accurate land-state information for a variety of land-surface applications, such as flood or drought forecasting. ERA5-Land uses the Carbon Hydrology-Tiled ECMWF Scheme for Surface Exchanges over Land (CHTESSEL) as the model [34]. The model has made significant improvements in its structure, taking into account the climatic seasonality of vegetation and using a new parametric method to calculate ET from bare-soil surfaces to better simulate ET over land. ERA5-Land provides global-scale ET data at 1 h intervals from January 1979 to the present. In this paper, the total ET data was used, the temporal resolution was increased from the hourly scale to the monthly scale, and the spatial resolution was increased from $0.1° \times 0.1°$ to $0.25° \times 0.25°$ to match other datasets.

#### 2.1.2. Global Land Data Assimilation System (GLDAS) ET

GLDAS was developed in collaboration with NASA, the National Oceanic and Atmospheric Administration, and the National Center for Environmental Prediction [35]. GLDAS uses advanced land-surface-modeling technology to generate the optimal surface state and flux field by combining satellite and ground-observation data, to generate different products in different land-surface models, but among the many land-surface model products, only Noah has been updated so far. This paper uses GLDAS-Noah product 2.1 land-ET data with a spatial resolution of $0.25° \times 0.25°$. We then added up the three-hour intervals to the monthly scale to match it with the other datasets.

### 2.1.3. The Second Modern-Era Retrospective Analysis for Research and Applications (MERRA-2) ET

MERRA-2 is a re-analysis dataset that has been updated since 1979 and that combines a large amount of satellite data. It combines satellite and traditional weather observations with simulated atmospheric processes to achieve the best simulation of the state of the Earth system [36]. MERRA-2 uses a catchment-surface model and a three-dimensional variable data assimilation scheme to provide hydrologic variables [37,38]. MERRA-2 provides data at 0.625° × 0.5° spatial resolution and hour-scale temporal resolution. This paper increased the MERRA-2 ET dataset from hourly to monthly in time scale and changes it to 0.25° × 0.25° using nearest-neighbor resampling to match the other input datasets.

### 2.1.4. Eddy Covariance (EC) ET

Flux towers based on Eddy Covariance (EC) technology are commonly used to monitor the dynamics of carbon dioxide ($CO_2$), water, and energy exchange between the biosphere and the atmosphere [39]. To verify the fusion dataset, this paper obtained the latent heat flux (LE) of 184 flux towers from the website (https://fluxnet.org/, accessed on 1 September 2023). The FLUXNET2015 dataset contains at least one year's data for each flux tower from 1992 to 2014 (Table S1). This paper directly uses the monthly flux data of each station and selects data whose quality evaluation is greater than or equal to 0.5 as for valid data. The conversion of latent heat flux from surface observation to ET is calculated using the following formula:

$$\mathrm{ET} = \frac{LE}{\lambda} \qquad (1)$$

In the formula, $\lambda$ has a fixed value of 2.45 MJ·kg$^{-1}$. The flux towers selected in this paper are located within ten different vegetation classifications (Figure 1).

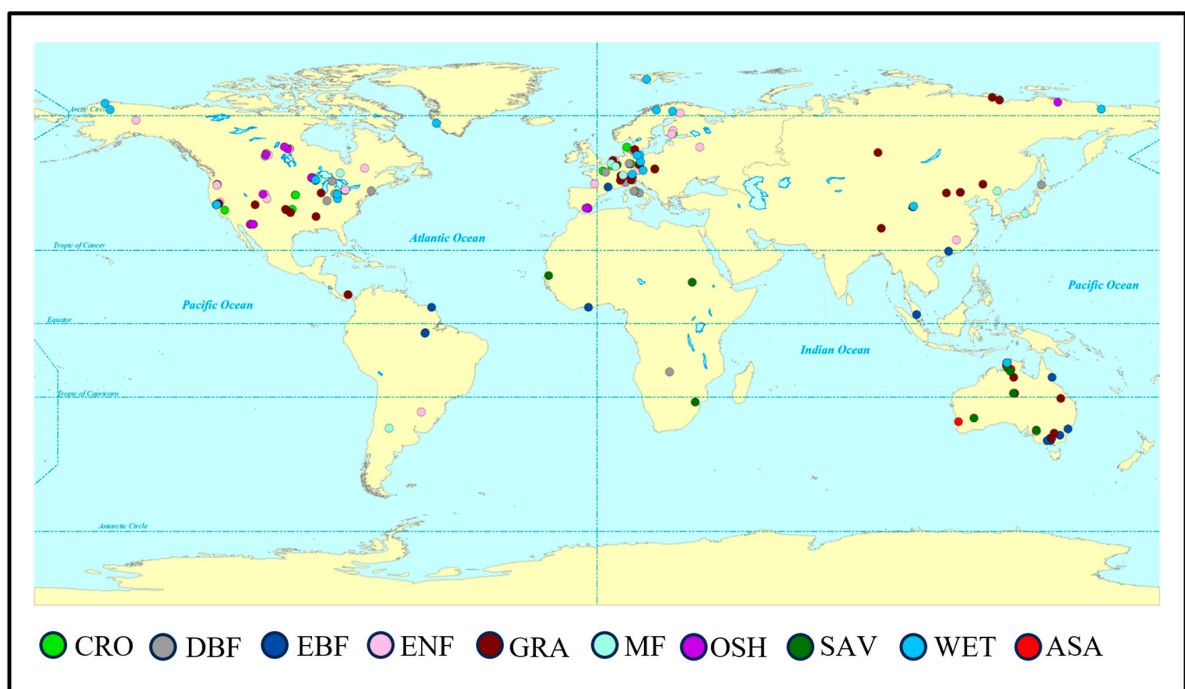

**Figure 1.** Spatial distribution of 184 in situ flux EC sites across the world. These sites include 19 cropland (CRO) sites, 26 deciduous broad-leaf forest (DBF) sites, 13 evergreen broad-leaf forest (EBF) sites, 43 evergreen needle-leaf forest (ENF) sites, 37 grassland (GRA) sites, 9 mixed forest (MF) sites, 12 open shrubland (OSH) sites, 8 savanna (SAV) sites, 19 permanent wetland (WET) sites and 6 woody savanna (ASA sites.

*2.2. Methods*

2.2.1. Uncertainty Analysis Based on the TCH Method

The TCH method does not need a measured value as a reference, but it needs three or more datasets to evaluate the uncertainty of its dataset. And no matter which dataset is used as a reference, its uncertainty is not affected [40]. In this paper, the TCH method was used to estimate the uncertainties on a monthly scale for each ET dataset, and also to calculate the uncertainties of 3 ET products for each of the 10 vegetation classifications, separately. The steps for calculating uncertainty by the TCH method are as follows:

For N kinds of time series, ET datasets can be expressed as:

$$ET_i = ET_{true} + \varepsilon_i, I = 1, 2, \ldots, N \tag{2}$$

where N is the number of ET datasets, which in this paper is 3. $ET_i$ is the i-th ET dataset, $ET_{true}$ is the true ET value, and $\varepsilon_i$ is the error between the i-th dataset and the true value.

Any dataset is selected as the reference dataset, and the difference between other datasets and the reference dataset is expressed as:

$$y_i = ET_i - ET_{ref} = \varepsilon_i - \varepsilon_{ref}, \text{€} = 1, 2, \ldots, N - 1 \tag{3}$$

where $ET_{ref}$ is any reference ET dataset, and $\varepsilon_{ref}$ is the error of the time series between the reference dataset and the truth value. Y is the matrix formed by N−1 difference values:

$$Y = \begin{bmatrix} y_{11} & \cdots & y_{1(N-1)} \\ \vdots & \ddots & \vdots \\ y_{M1} & \cdots & y_{M(N-1)} \end{bmatrix} \tag{4}$$

where, *M* is the number of time series data sets from each ET dataset. Since the monthly scale is used as input in this paper, *M* is 252.

The covariance of the difference sequence matrix *Y* is expressed as:

$$S = \text{cov}(Y) = \begin{bmatrix} s_{11} & \cdots & s_{1(N-1)} \\ \vdots & \ddots & \vdots \\ s_{(N-1)1} & \cdots & s_{(N-1)(N-1)} \end{bmatrix} \tag{5}$$

where *S* is the covariance matrix, cov(·) is the covariance operator, $s_{ij} = s_{ji}$ (i, j = 1, 2, ..., N − 1) is the variance (i = j) or covariance (i ≠ j) between $y_i$ and $y_j$.

Introducing an N × N covariance matrix R ($R_{ij} = \text{cov}(\varepsilon_i, \varepsilon_j)$, ε can be obtained from Equation (2) (R is a symmetric matrix).

The relationship between R and S is expressed as:

$$S = J \cdot R \cdot J^T \tag{6}$$

The square root of the diagonal value of the R($\{\sigma_{ii}\}$i=1, 2, ..., N) matrix ($\sigma_i$) is considered to be the TCH uncertainty of the i-th ET product and also the standard deviation of the i-th month-scale ET time series dataset.

2.2.2. Fusion Method

After obtaining product uncertainty $\sigma_i$ by the TCH method, data fusion was carried out. First, we assume that the ET time series dataset on each monthly scale is normally distributed, and its probability density function is:

$$P(ET_i|ET_{true}) = \frac{1}{\sigma_i\sqrt{2\pi}} \exp[-\frac{\varepsilon_i^2}{2\sigma_i^2}] = L(ET_{true}|ET_i) \tag{7}$$

where, P (·) is the probability density function and L (·) is the likelihood function. $ET_i$ is the i-th ET dataset, $ET_{true}$ is the true ET value, and $\varepsilon_i$ is the error between the i-th dataset and the true value.

Similarly, the ET time series dataset on the j-th monthly scale can be expressed as:

$$P(ET_j|ET_{true}) = \frac{1}{\sigma_j\sqrt{2\pi}}\exp[-\frac{\varepsilon_j^2}{2\sigma_j^2}] = L(ET_{true}|ET_j) \tag{8}$$

where $ET_j$ is the j-th ET dataset, $\varepsilon_j$ is the error between the j-th dataset and the true value, and $\sigma_j$ is the standard deviation of the j-th monthly scale ET time series dataset.

The maximum likelihood value of $ET_{true}$ can be expressed as the maximum of its joint probability distribution above:

$$\max L(ET_{true}|ET_i, ET_j) = P(ET_i|ET_{true})P(ET_j|ET_{true}) = \frac{1}{2\pi\sigma_i\sigma_j}\exp[-\frac{\varepsilon_i^2}{2\sigma_i^2} - \frac{\varepsilon_j^2}{2\sigma_j^2}] \tag{9}$$

Taking the logarithm of both sides of Formula (9), we obtain:

$$\ln L(ET_{true}|ET_i, ET_j) = -\ln(2\pi\sigma_i\sigma_j) - \frac{\varepsilon_i^2}{2\sigma_i^2} - \frac{\varepsilon_j^2}{2\sigma_j^2} \tag{10}$$

By taking the partial derivative of $ET_{true}$ on both sides of Formula (10), we obtain:

$$\frac{\partial lnL}{\partial ET_{true}} = -\frac{ET_i - ET_{true}}{\sigma_i^2} - \frac{-ET_{true}}{\sigma_j^2} \tag{11}$$

Set Formula (11) equal to 0, we obtain:

$$ET_{true} = \frac{\sigma_i^2}{\sigma_i^2 + \sigma_j^2}ET_i + \frac{\sigma_j^2}{\sigma_i^2 + \sigma_j^2}ET_j \tag{12}$$

Formula (12) can be expressed as $ET_{true} = v_iET_i + v_jET_j$, and the fusion weight of each ET dataset is:

$$v_i = \frac{\sigma_i^2}{\sigma_i^2 + \sigma_j^2} v_j = \frac{\sigma_j^2}{\sigma_i^2 + \sigma_j^2} \tag{13}$$

In the formula, $v_i$ and $v_j$, respectively, represent the fusion weight of the *i*-th and *j*-th kind of dataset.

For *N* datasets, the fusion weight can be expressed as:

$$v_k = \frac{\prod_{i=1, i\neq k}^{N}\sigma_k^2}{\sum_{k=1}^{N}(\prod_{i=1, i\neq k}^{N}\sigma_k^2)} \tag{14}$$

where N is the number of ET datasets and $v_k$ is the fusion weight of the *k*-th kind of dataset.

Using its fusion weight to calculate the month-scale ET after fusion by pixel:

$$ET_m = \sum_{k=1}^{N} v_kET_k \tag{15}$$

where, $ET_m$ is the fused ET dataset, and $ET_k$ is the *k*-th ET dataset.

### 2.2.3. The Verification Method of the Fused Product

In this paper, EC latent heat flux data was converted into ET data to evaluate and validate the fused ET. Pearson correlation coefficient (r), root mean square deviation (RMSE), and relative deviation percentage (BIAS, %) were used to evaluate and verify the products. r quantifies the degree of linear correlation between product values and flux observations.

RMSE reflects the degree to which product data deviates from observed values. BIAS is the degree to which product data is overestimated (positive) or underestimated (negative) relative to observations. The expressions of r, RMSE and BIAS are:

$$r = \frac{\sum_{i=1}^{n}(M_i - \overline{M})}{\sqrt{\sum_{i=1}^{n}(M_i - \overline{M})^2}\sqrt{\sum_{i=1}^{n}\left(ref_i - \overline{ref}\right)^2}} \tag{16}$$

$$RMSE = \sqrt{\frac{\sum_{i=1}^{n}(M_i - ref_i)^2}{n}} \tag{17}$$

$$BIAS = \frac{\sum_{i=1}^{n}(M_i - ref_i)}{\sum_{i=1}^{n} ref_i} \times 100 \tag{18}$$

In the formula, $n$ represents the sample size, $M_i$ represents multiple sets of ET re-analysis data at time $i$, and $ref_i$ is used as reference data at time $i$, which is EC data in this paper. $\overline{M}$ and $\overline{ref}$ represent the average of $M_i$ and $ref_i$.

## 3. Results and Discussion

### 3.1. Spatial and Temporal Distribution of the Three ET Products

The spatial distribution of the annual mean ET of the three datasets shows good consistency (Figure 2). The regions with high land ET were concentrated near the equator, whose climate is humid, including the Amazon Plain in northern South America, the Congo Basin in central Africa, and the junction of Asia and Oceania. The average annual rainfall in these areas usually exceeds 1000 mm/year. Extremely low land ET is concentrated in very arid deserts and permafrost regions, including the Sahara Desert and the Arabian Desert in northern Africa; the deserts of Taklimakan, Turkey, Iran, and India in Central Asia; and the permafrost regions of northern North America and Eurasia, whose average annual rainfall is usually less than 200 mm/year. MERRA-2 ET was significantly higher than ERA5-Land and GLDAS-Noah in the 50°N–90°N region, and ERA5-Land ET was significantly higher than the other two products in the western region of South America. At the same time, MERRA-2 resampling assigns values to some water bodies; this part is not considered in this paper, so it has no influence. In general, MERRA-2 has the highest annual ET value, while ERA5-Land and GLDAS-Noah have little difference in ET value. The overall trend of the three ET products in terms of latitude was consistent (Figure 2d).

Figure 2e describes the changing trend of the annual mean value of three ET products from 2002–2022. The annual average of MERRA-2 ET is about 10–20% higher than that of ERA5-Land and GLDAS-Noah. The variation trend of the annual mean value of the three products is consistent but the degree of change is different. MERRA-2 and GLDAS-Noah have a remarkable yearly change, while ERA5-Land has a gentle change. In summary, the three ET products have significant differences in some regions and years, indicating that the estimation of land ET in some regions or years has great uncertainty, but the overall trend is relatively consistent.

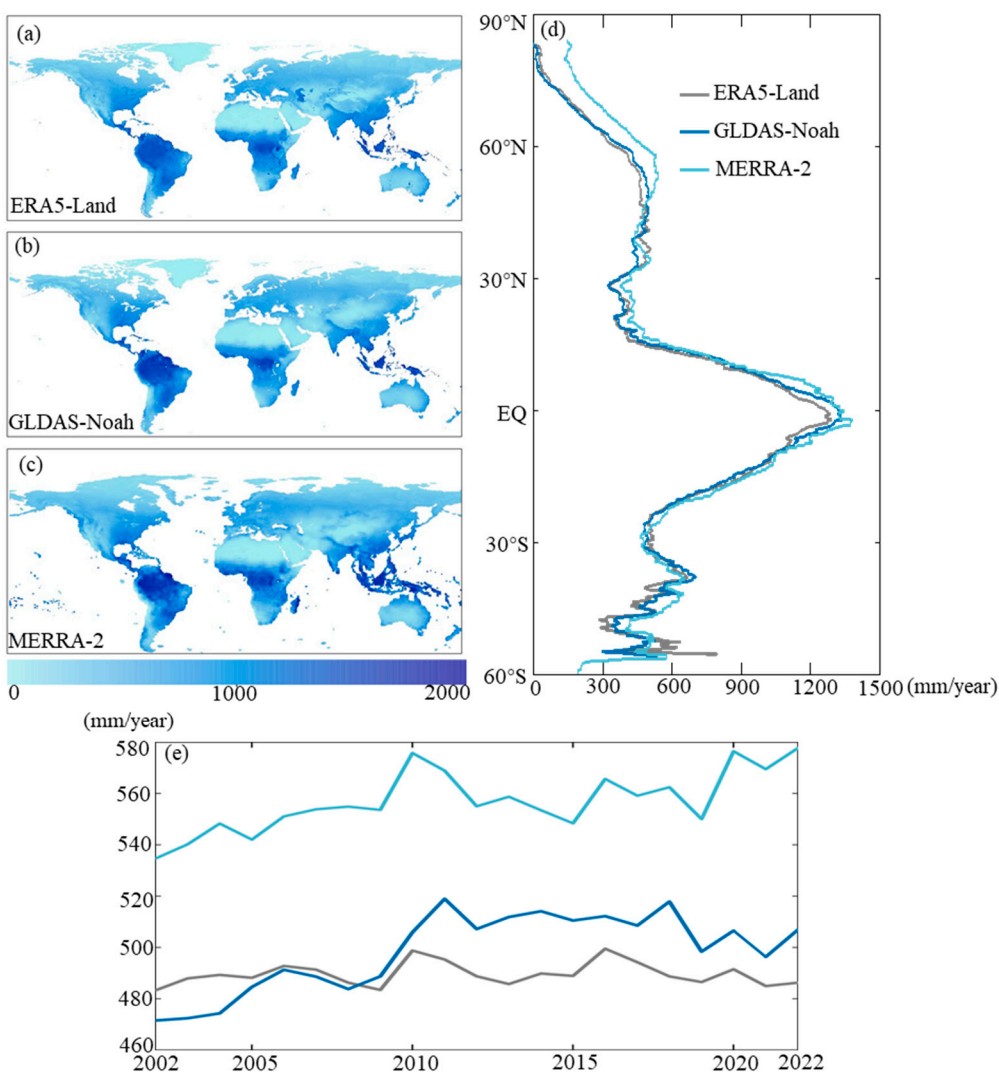

**Figure 2.** (**a**–**c**) Spatial distribution of annual mean land ET for the period 2002–2022 (unit: mm/year). (**d**) Latitudinal distribution of annual mean land ET, (**e**) time series (2002–2022) distribution of annual mean land ET from three ET products.

### 3.2. The Uncertainty of the Three Products

In general, the distribution of global uncertainty (UC) of the three ET products is similar (Figure 3). ERA5-Land product has a lower average UC overall, but higher UC values in the Tibetan Plateau and southern India (Figure 3a). Since the specific ERA5-Land algorithm is not public, it is difficult to analyze the cause of uncertainty. While considering that ERA5-Land mainly combines model data with observational data from around the world [41], the above situation may be related to the lack of weather stations in this region, resulting in sparse and unrepresentative ET data. Wang et al., (2012) showed that the GLDAS product had the best precipitation performance in the Tibetan Plateau region. Precipitation and air temperature in the MERRA product have the highest correlation with the site data in the monthly scale [42]. Precipitation and temperature are the key factors driving evapotranspiration, and their research indirectly indicated that the MERRA and GLDAS products have low uncertainty over the Tibetan Plateau. In the TCH method, the lower UC of MERRA-2 and GLDAS-Noah will lead to a higher UC for ERA5-Land, which is also one of the factors that may lead to the high UC of ERA5-Land in this paper. Of course, the complex topography and climate variability of the Tibetan Plateau will also lead to high UC in the estimation of evapotranspiration. The UC of GLDAS-Noah is fairly balanced across the globe, with small UC in most regions and high UC only in south-east Asia, the

south-eastern coastal regions of North America, and the southern Sahara Desert (Figure 3b). The MERRA-2 product has a slightly higher overall average UC and the high UC regions are mainly distributed in the interior of the Arctic Circle, north-central South America, south-east Asia, and most of Africa, but it has smaller UC in arid regions such as the Sahara Desert (Figure 3c). MERRA-2 shows high UC, which may be due to its greater emphasis on assimilation resulting from the use of satellite observation data. Especially in tropical regions, due to cloudy and rainy weather conditions, it is impossible to obtain effective and accurate satellite data, resulting in a large difference in the spatial distribution of UC of MERRA-2 products. Of course, the performance of land-ET products also varies according to different estimation models in different climate zones. Feng et al., (2018) analyzed the correlation between land ET estimated based on the Budyko hypothesis and reanalyzed ET products, mainly because the variance of MERRA-2 is larger than that of other reanalyzed products, indicating that MERRA-2 has major problems in describing the annual change and long-term trend of land ET in China. In addition, the study shows that MERRA-2 also has great uncertainty in semi-arid, semi-humid, and humid regions [43].

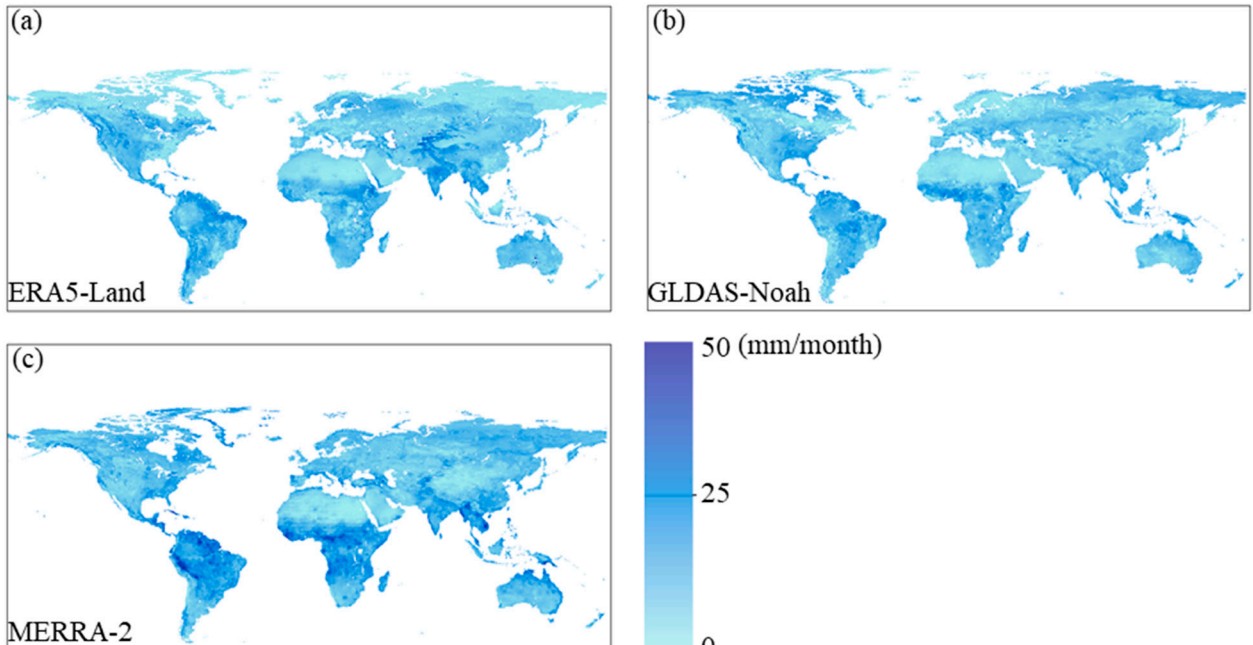

**Figure 3.** Spatial distribution of uncertainty of the three ET products (**a–c**).

We calculated the UC of ET for different vegetation classifications (Figure 4). The differences between different vegetation classifications can be represented by the UC of different products. The median UC was less than 10 mm/month for every vegetation classification except EBF, SAV, and ASA. The average UC values of DBF (9.41 mm/month), MF (9.18 mm/month), and WET (9.12 mm/month) are relatively small. The average UC values of EBF (13.96 mm/month), SAV (11.08 mm/month), and ASA (12.12 mm/month) were higher. For distribution, DBF, MF, and OSH are more concentrated, while the UC of EBF, ENF, and WAS show large spatial differences.

Among the three ET products of different vegetation classifications, GLDAS-Noah shows the lowest UC mean value, ranging from 7.09 mm/month (OSH) to 12.49 mm/month (EBF). However, GLDAS-Noah has the largest UC in CRO (11.29 mm/month) and GRA (9.75 mm/month). The research results of Wang et al., (2011) also pointed out that the GLDAS product also has relatively high errors in farmland areas, and they attributed this to the uncertainty driving the product as an inherent part of the assimilation process [44,45]. ERA5-Land shows minimal UC in CRO, EBF, GRA, MF, and WET. Because each ET estimation model has a different sensitivity to different vegetation classifications,

the existing land-ET products each have their unique advantages and limitations for specific vegetation classifications. This paper finds that ERA5-Land performs well under vegetation classifications with sufficient moisture, which is consistent with the result of Munoz-Sabater et al., (2021) that ERA5-Land overestimates ET in arid areas due to ET stress caused by strong and dry air in the ERA5-Land model [46]. In the forest regions of EBF, ENF, and MF, all three datasets exhibit relatively large UC. Of the three products, MERRA-2 performs the worst (average 13.68 mm/month), followed by ERA5-Land (average 10.03 mm/month). GLDAS-Noah has the lowest UC among these regions (average 9.39 mm/month). However, Khan et al., (2018) showed the greatest absolute and relative uncertainty in calculating GLDAS ET products based on TC methods in different forest areas [47]. Gomis-Cebolla et al., (2019) believed that due to the uncertainty of radiation propagation in different surface models of ET products, for example, MERRA-2 tends to overestimate incident radiation and net radiation, while GLDAS tends to underestimate incident radiation and overestimate daily radiation, resulting in greater uncertainty in estimating ET in forest areas [48]. Different ET products have different models for estimating different vegetation classifications, complex terrain, and nesting among various vegetation, resulting in highly inconsistent results. This is consistent with the results of Zhu et al., (2022), who performed a comprehensive evaluation of 16 global ET products which shows that it is difficult or even impossible to identify the best ET products in all aspects [19].

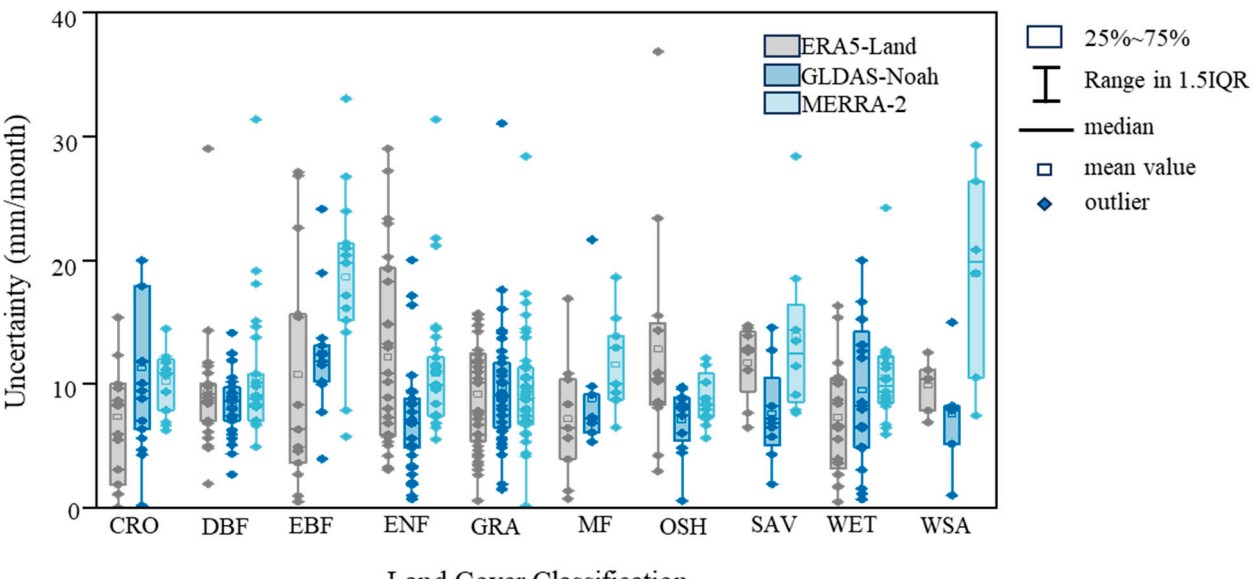

**Figure 4.** Statistics on the distribution of ET uncertainty for different vegetation classifications. Cropland (CRO), deciduous broad-leaf forest (DBF), evergreen broad-leaf forest (EBF), evergreen needle-leaf forest (ENF), grassland (GRA), mixed forest (MF), open shrubland (OSH), savanna (SAV), permanent wetland (WET), and woody savanna (ASA).

*3.3. The Weight of the Three Products*

Figure 5a–c shows the fusion weights on each grid of the three ET products. This weight can represent the confidence of a single grid and the degree to which each product contributes to the fusion product. The higher the weight, the better the product performs in the region. By calculating the global average weight of each ET product, it is found that ERA5-Land has the largest weight (0.42), followed by GLDAS-Noah (0.36) and MERRA-2 (0.22).

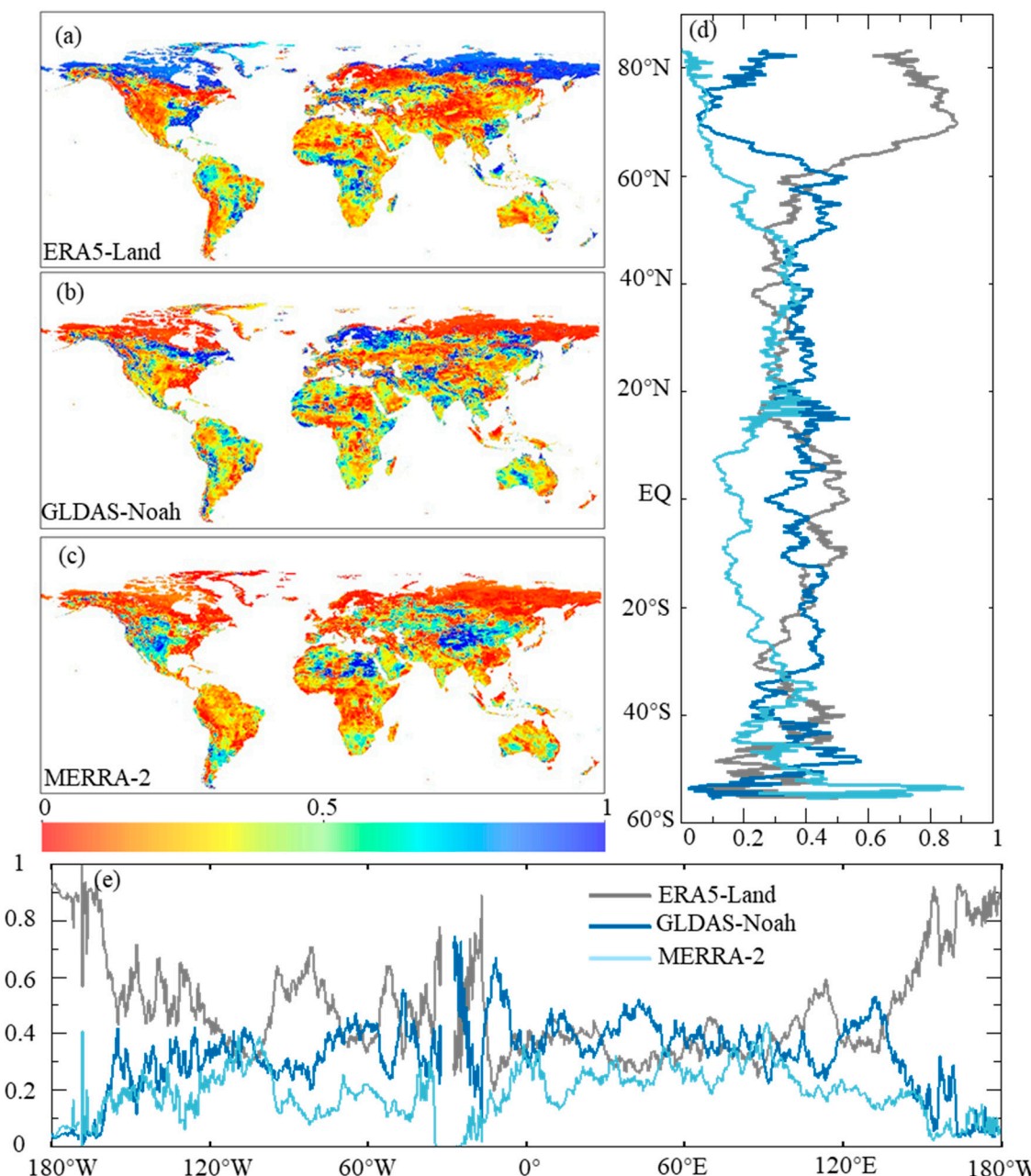

**Figure 5.** (**a**–**c**) Spatial distribution of weights, (**d**) latitudinal distribution of weights, and (**e**) longitudinal distribution of weights.

Within the Arctic Circle, near the equator, in the south-eastern United States, and in south-eastern China, the weight of ERA5-Land is greater. The weight of ERA5-Land in these regions is above 0.8, while the weight of GLDAS-Noah and MERRA-2 in these regions is relatively small, indicating that the integration contribution of ERA5-Land in these regions is superior. GLDAS-Noah has a greater weight in northern North America, northern Europe, and the Middle East, while MERRA-2 has a greater weight in the central United States, African deserts, and the Tibetan Plateau.

The weights of each product at different latitudes are described as shown in Figure 5d. Obviously, in the high latitudes of the Northern Hemisphere, the weight difference between the three products is the largest, with ERA5-Land having the greatest weight. In the 20°N–40°N region, the weights of the three products are the most similar. Near the equator MERRA-2 has less weight than the other two products; further, ERA5-Land has slightly

more weight than GLDAS-Noah. South of 40°S in the Southern Hemisphere, MERRA-2 gradually gained more weight than the other two datasets. The weights of each product at different longitudes are depicted as shown in Figure 5e. In the 180°W–30°W and 140°E–180°E regions, ERA5-Land has the most weight overall, followed by GLDAS-Noah. At 0°–100°E, the weight of the three products is not much different. In contrast to the previous fusion method, the weight is distributed in the reciprocal of the number of input products (for example, if the input product is 3, the weight is distributed at 3.33) [6]. The minimum weight of each ET product in this paper is almost zero, and the maximum is close to one. Such a large and violent fluctuation may be caused by the fact that the TCH method does not rely on the reference product to calculate the weight, and cannot obtain a value as a reference line, which will amplify the degree of uncertainty in determining the fusion weight, resulting in violent fluctuations of the weight.

### 3.4. Evaluation of Products after Fusion

Figure 6 shows the scatter plots of observation data from multiple ET products and sites on a monthly scale. Most of the points are concentrated above the 1:1 line, and the BIAS is positive, which shows that the land ET of these ET products is overestimated to a certain extent. Among the four ET products, TCH and ERA5-Land had the highest r within site observation data, reaching 0.77, followed by GLDAS (0.75) and MERRA-2 (0.73). For RMSE, ERA5-Land is the smallest (26.79 mm), followed by TCH (27.94 mm), GLDAS-Noah, and MERRA-2 at 29.99 mm and 33.79 mm, respectively. For BIAS absolute values the order is ERA5-Land < GLDAS-Noah < TCH < MERRA-2. In summary, the deviation between ERA5-Land and site observation data is significantly smaller than that of other ET data products, indicating its high accuracy. But ERA5-Land combines model data with observations from around the world to form a globally complete, consistent dataset. ERA5-Land may have a high autocorrelation with the site observation data. Of course, TCH is the fusion of three products, so it should also have a certain degree of autocorrelation with the site observation data, but compared with ERA5-Land, the autocorrelation will undoubtedly be much smaller. If the performance of ERA5-Land is excluded, the TCH fusion dataset has the highest correlation coefficient with site observation data. At the same time, its RMSE is the smallest, and its BIAS absolute value is almost the same as the minimum value, indicating that TCH performs better than GLDAS-Noah, which also uses data assimilation technology and combines ground observation and satellite remote-sensing data to drive surface model development, and it also outperforms MERRA-2, which emphasizes the use of satellite observation data.

Previous studies have shown that there is a close relationship between the quality of land-ET products and vegetation [49]. To further evaluate the performance of ET products under different vegetation classifications, ET products under different vegetation classifications were validated. Table 2 quantitatively describes the performance of ET products at ten vegetation-classification sites on a monthly scale from three indicators: r, RMSE, and BIAS. The results show that no single ET product performed best under all vegetation classifications. Previous studies have also confirmed this point. For example, Majozi et al., (2017) evaluated the accuracy of four ET products in two typical ecological zones in South Africa and found that no ET product performed best in any two regions [5]. Kim et al., (2012) found that the performance of MODIS ET products in Asian forest areas is better than that of other vegetation cover classifications [50]. Ershadi et al., (2014) also pointed out that the performance of ET models in Europe and North America is different in a certain vegetation classification, and the model corresponding to better performance of each vegetation classification is variable [51]. TCH and ERA5-Land have different emphases in different vegetation classifications, which is much better than GLDAS-Noah and MERRA-2. Although TCH performs best under only five classifications of planting, it performs equally well under the other five classifications of planting.

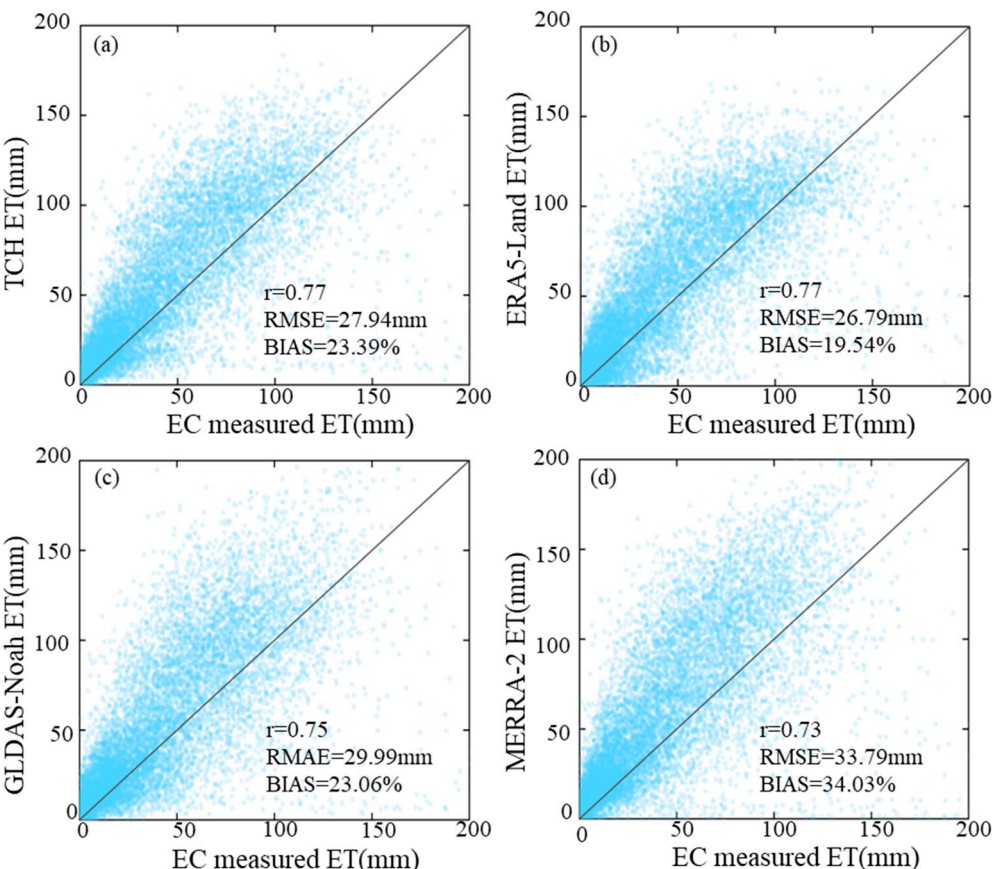

**Figure 6.** (**a**–**d**) Scatterplots of monthly EC, measured ET, and ET from different products. The 1:1 line is plotted in black. The correlation coefficients (r) have passed the 5% significance test.

**Table 2.** The verification results including r, RMSE, and BIAS (%) between monthly EC measured ET and monthly ET from different products in different vegetation classifications. Values in bold indicate the highest quality.

| Vegetation Classifica-tions | TCH | | | ERA5-Land | | | GLDAS-Noah | | | MERRA-2 | | |
|---|---|---|---|---|---|---|---|---|---|---|---|---|
| | r | RMSE | BIAS | r | RMSE | BIAS | r | RMSE | BIAS | r | RMSE | BIAS |
| CRO | 0.66 | 34.45 | 20.26 | **0.70** | **32.02** | 20.59 | **0.70** | 36.48 | 26.21 | 0.60 | 37.84 | **19.64** |
| DBF | **0.82** | 29.40 | 41.27 | **0.82** | **28.79** | **41.24** | **0.82** | 30.70 | 42.01 | 0.76 | 33.39 | 43.61 |
| EBF | 0.76 | 29.44 | 16.98 | **0.79** | **25.45** | 16.06 | 0.71 | 32.30 | **15.21** | 0.73 | 40.36 | 23.27 |
| ENF | **0.80** | 24.56 | 24.54 | 0.77 | **23.49** | **15.49** | 0.78 | 27.30 | 26.36 | 0.75 | 33.25 | 47.71 |
| GRA | **0.90** | **18.82** | 23.47 | 0.84 | 20.64 | **13.77** | 0.83 | 22.22 | 17.05 | 0.88 | 23.67 | 33.64 |
| MF | 0.83 | 30.24 | 52.70 | 0.84 | **27.47** | 46.48 | 0.75 | 32.78 | **43.31** | **0.87** | 36.00 | 72.88 |
| OSH | **0.81** | **19.08** | 36.25 | 0.69 | 21.99 | **31.74** | 0.80 | 20.66 | 43.90 | 0.77 | 23.13 | 44.70 |
| SAV | **0.86** | **20.71** | 6.06 | 0.81 | 22.10 | −2.87 | 0.85 | 21.70 | 6.56 | 0.85 | 27.80 | 15.37 |
| WET | 0.46 | 46.69 | −2.02 | **0.53** | **42.79** | −3.98 | 0.47 | 47.01 | **−1.29** | 0.41 | 49.74 | 4.53 |
| ASA | **0.84** | **21.92** | **−6.41** | 0.83 | 22.75 | 7.55 | 0.83 | 22.84 | −7.69 | 0.79 | 31.41 | 10.43 |

For GRA, OSH, SAV, and ASA flux sites, TCH has the highest r and lowest RMSE of any of the ET products. For DBF and ENF flux sites, TCH has the highest r. The BIAS absolute value of TCH is minimal only at ASA sites. TCH performs worse than at least one single ET product on every other site. Specifically, at CRO sites, TCH's r is 0.66, lower than ERA5-Land; RMSE is 34.45 mm/month, higher than ERA5-Land; and the absolute value of BIAS is greater than MERRA-2. For EBF sites, TCH's r is 0.76, RMSE is 29.44 mm/month,

and BIAS's absolute value is 16.98, all of which are worse than ERA5-Land. For MF sites, TCH's r is 0.83, lower than ERA5-Land and MERRA-2; RMSE is 30.24 mm/month, higher than ERA5-Land; and BIAS's absolute value is 52.70, higher than ERA5-Land and GLDAS-Noah. For WET sites, TCH's r is 0.46, lower than ERA5-Land and GLDAS-Noah; RMSE is 46.69 mm/month, higher than ERA5-Land; and BIAS absolute value is 2.02, higher than GLDAS-Noah. Overall, TCH, ERA5-Land, GLDAS-Noah, and MERRA-2 performed best in five (including ENF, GRA, OSH, SAV, and ASA), five (CRO, DBF, EBF, MF, and WET), zero, and zero vegetation classifications, respectively. The total performance of TCH and ERA5-Land is not much different. However, after removing ERA5-Land, in a comparison of TCH, GLDAS-Noah, and MERRA-2, excluding CRO and WET sites, TCH performs slightly worse than GLDAS-Noah, and the comprehensive performance of other vegetation-classification sites is better than GLDAS-Noah and MERRA-2. TCH is not the best in all aspects of all vegetation classifications; however, it avoids being the worst performer in any vegetation classification. It can be shown that the new ET product fused using the TCH method greatly reduces the uncertainty of ET estimation. After comprehensive evaluation, TCH can be used for most vegetation types.

### 3.5. Spatial and Temporal Distributions of the Fusion Product

Figure 7a describes the spatial distribution of the average annual land ET of the fusion product TCH, and finds that it is consistent with the average annual land-ET distribution of the three input datasets mentioned above. There are some differences in the global average annual ET values of the four products, among which the MERRA-2 product is the highest (557.14 mm/year) and the TCH product is the second highest (508.73 mm/year), followed by GLDAS-Noah (498.61 mm/year). ERA5-Land is the smallest (489.62 mm/year). After fusion, the difference between the TCH product and products is smaller than the difference between each product before fusion, which reduces the uncertainty to a certain extent. From the latitude distribution, the overall trend of ET of the TCH product was the same as that of the three input datasets (Figure 7b). MERRA-2 is significantly higher than TCH in the 60°N–90°N region. Compared with TCH, the measurements of MERRA-2 and GLDAS-Noah are significantly higher in very wet regions near the equator, and generally consistent elsewhere. It shows that the TCH product successfully captures the spatial differences of land-ET products in a long time series.

Figure 7c describes the annual mean values of multiple ET products and their changing trends during 2002–2022. In addition to ERA5-Land, the average annual ET of other ET products increased significantly. There are great differences in the trend of annual ET of the four kinds of products. GLDAS-Noah shows the highest trend of ET increase, and its value (1.78 mm/year) is about 1.3 times that of MERRA-2 (1.36 mm/year), 1.7 times that of TCH (1.07 mm/year), and 35.6 times that of ERA5-Land (0.05 mm/year). It can be shown that ERA5-Land has great uncertainty in the change trend of long-term time series, and the integrated product TCH can effectively avoid the influence of the uncertainty of ERA5-Land on the whole, and effectively capture the difference in the time-change trend of land-ET products.

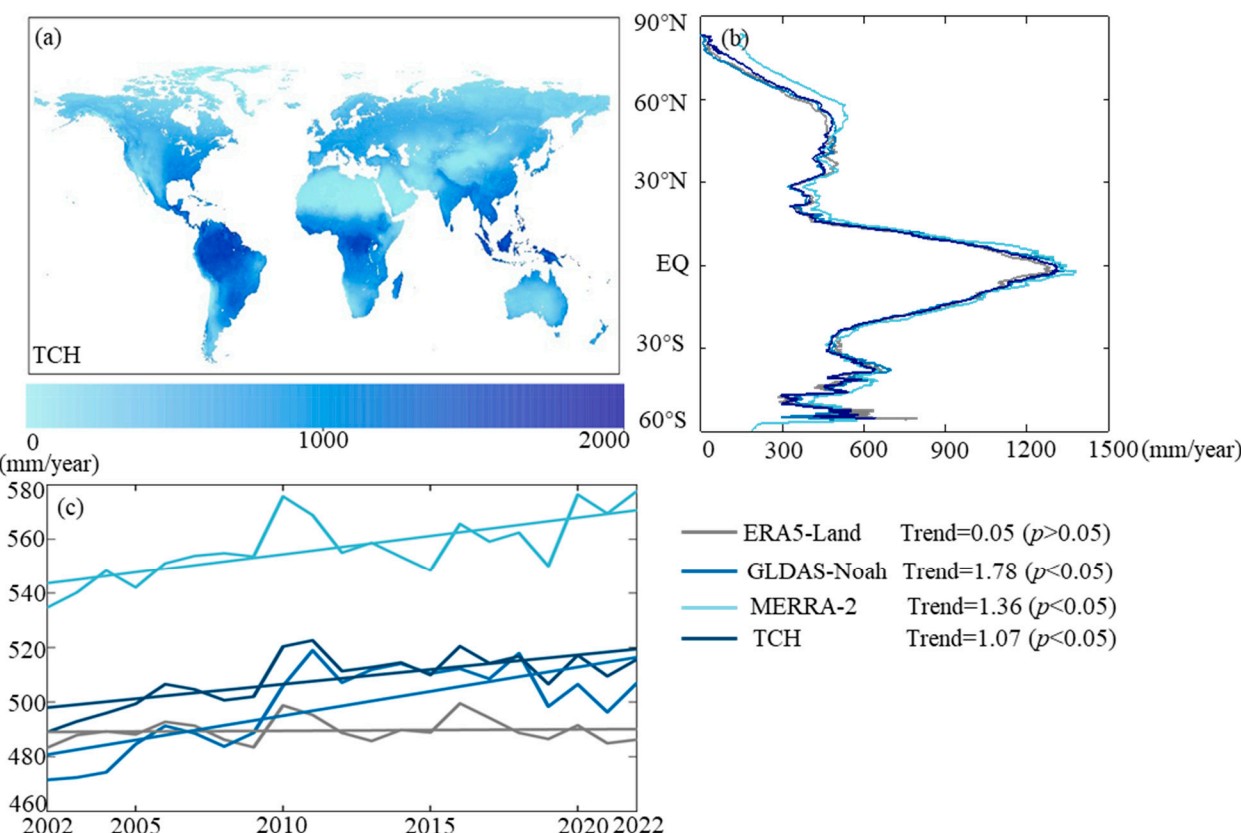

**Figure 7.** (**a**) Spatial distribution of annual mean land ET for the period 2002–2022 of TCH (unit: mm/year). (**b**) Latitudinal distribution of annual mean land ET. (**c**) Time series (2002–2022) distribution of annual mean land ET and tread of four ET products.

### 3.6. Spatial Distribution of the Linear Trend of Four ET Products

Figure 8 depicts the changing trend of ET for multiple products from 2002 to 2022. The fusion ET product TCH shows a significant decrease in south-eastern South America and southwestern parts of Africa, while ET increases in almost all other regions, including eastern North America, north-eastern South America, western Europe, north-central Africa, southern Asia, and south-eastern Oceania. In tropical regions near the equator, except for MERRA-2, which shows a significant downward trend in South America, other products show an upward trend in this range, among which GLDAS-Noah increases most strongly, while ERA5-Land shows a less significant increase trend. After evaluating a variety of ET products, Wenbin et al., (2022) found that most ET products showed an increasing trend of ET in the Amazon Basin of South America [19]. Compared with the results presented in this paper, the results show that MERRA-2 has a large uncertainty in the trend of ET change in South America. In the vicinity of the Congo Basin in Central Africa, ERA5-Land ET shows a decreasing trend, while other products show an increasing trend. Burnett et al., (2020) found by analyzing environmental data that the Congo Basin has become sunnier and less humid in recent years [52]. This result is consistent with an expected increase in ET. It shows that TCH, GLDAS-Noah, and MERRA-2 are close to ET trends in the Congo Basin in Africa, while this paper and other studies clearly show that MERRA-2 ET has the largest trend of increase in the Congo Basin [52,53]. ET increases can be found in all products in eastern North America, western Europe, and northern Asia, as well as in south-east Asia. In the northern coastal regions of Oceania, TCH, GLDAS-Noah, and ERA5-Land ET also show an increasing trend, while MERRA-2 ET shows a clear opposite trend. Dong et al., (2016) showed that the increase in temperature and rainfall will lead to a significant upward trend of land ET, due to the warming of the climate in recent years [54]. This is consistent with the upward trend of ET of all products in this paper, among which MEERA-2 ET has

a larger upward trend, but the overall upward trend of ERA5-Land ET is not significant. The ET trend of the integrated product TCH is more balanced, which is neither as "radical" as MERRA-2 nor as "conservative" as ERA5-Land. It effectively captures the abnormal trend of various products. For example, the decrease of ET within Central Africa in ERA5-Land, the increase of ET within northeast North America in GLDAS-Noah, or the drastic decrease of ET within SouthAmerica in MERRA-2 products, etc. TCH effectively reduces the uncertainty of the time-change trend of ET in spatial distributions.

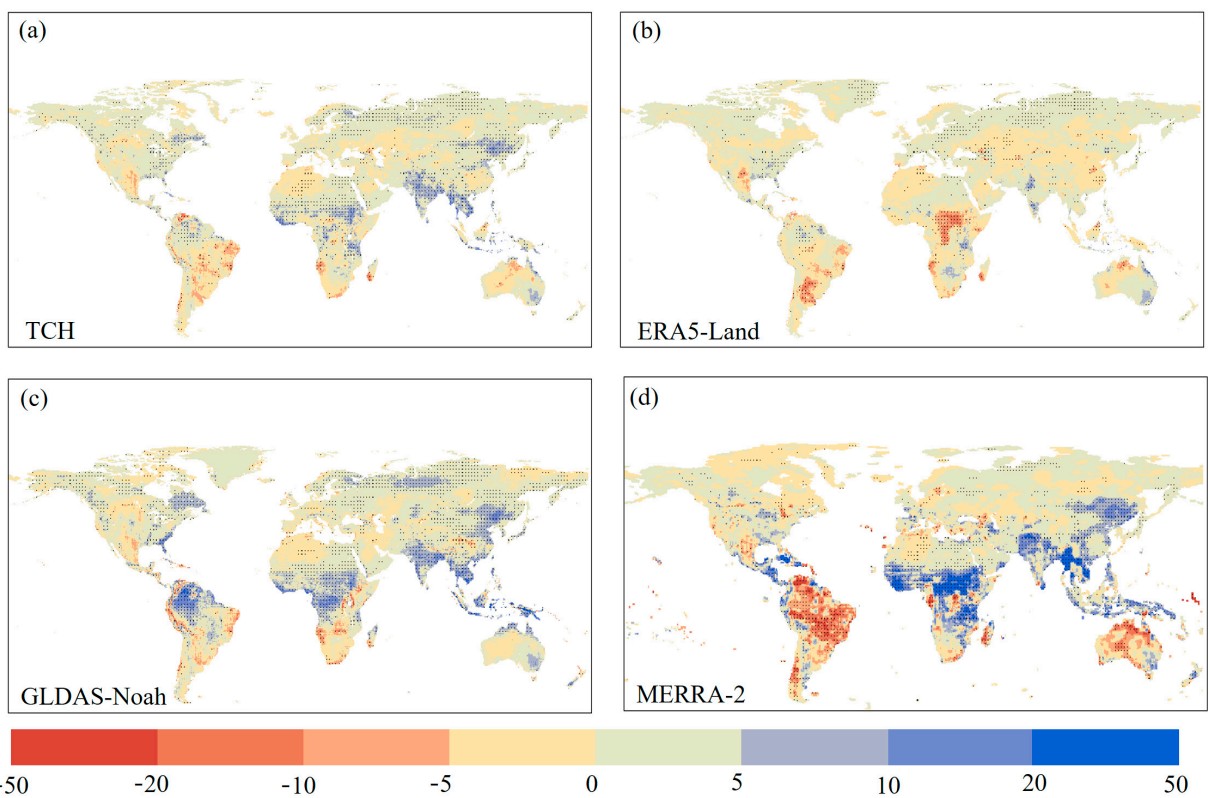

**Figure 8.** Spatial distribution of linear trends of land ET for the period 2002–2022 (unit: mm/year). Little points indicate that regions have passed the 5% significance test.

The uncertainty in the fusion of ET products is driven by many factors such as input error, scaling effect, and fusion algorithm. Input errors are derived from individual ET products and EC ground-observation data. The algorithm of each ET product has certain defects, which inevitably makes the ET product have certain uncertainties. The EC observation data determines the accuracy of the fused ET product, as it is often considered the standard reference data value used to evaluate the product [55,56]. But, due to problems such as unclosed energy, EC observation data has 5–20% error [57]. When the ET product is evaluated, the uncertainty of EC data will lead to the uncertainty of the ET product. The uncertainty caused by the scaling effect is caused by the mismatch between the spatial resolution of the ET product and the radiation range of the flux tower. Specifically in this paper, the spatial resolution of the four kinds of monthly scale ET products is uniformly $0.25° \times 0.25°$. However, the footprint of EC observations is usually limited to a few kilometers from the flux tower [58]. The uncertainty in the fusion algorithm is caused by the differential calculation of the weight of each product. In addition, many other factors can lead to uncertainty. For example, in some common models for estimating ET, the ET estimates are very dependent on the input data, so that the fused data is limited to the maximum value range of the input data. However, if the input data are all overestimated or underestimated, the fused dataset will also be overestimated or underestimated; even higher uncertainty than the least overestimated/underestimated datasets before the fusion.

Today's evaluation and integration methods are not enough to solve the above problems at the same time, and they need to weigh both advantages and disadvantages to improve the accuracy of product performance. Similarly, the TCH method adopted in this paper only considers the uncertainty relationship between various products and does not discuss the differences in their internal algorithms and mechanisms. At the same time, the weight obtained by the TCH method has only one value in the long time series, and the change in time difference is not considered. Therefore, future studies need to combine surface-energy balance and water balance to determine the physical mechanism of uncertainty in various products and strengthen the precise quantification of ET products. At the same time, the time difference should be considered to realize the dynamic change of fusion weight in time.

## 4. Conclusions

In summary, we evaluated the uncertainty of three land ET datasets by the TCH method, and we fused the ET datasets with their uncertainty relationships as weights, and generated a long-series global monthly ET dataset with a spatial resolution of $0.25° \times 0.25°$ and a time span of 21 years.

The evaluation of ERA5-Land, GLDAS-Noah, and MERRA-2 ET products using the TCH method shows that ERA5-Land performs better in areas with sufficient water vapor, while MERRA-2 has lower uncertainty in arid areas. The performance of GLDAS-Noah is more balanced, and the performance is more prominent on shrubs, grasslands, and other low vegetation. It shows that different ET products have different degrees of adaptation to different vegetation types and climate regions, and it is difficult to simply use one ET product to estimate ET in all regions.

The performance of the ET products after fusion is more reliable, and the uncertainty of ET estimation is effectively reduced. Overall validation with FLUXNET data based on r, RMSE, and BIAS showed that, in all datasets, the fusion product and ERA5-Land performed differently in different vegetation classifications, and the overall performance was not much different, far better than GLDAS-Noah and MERRA-2. TCH had the best performance in estimating ET in ENF, GRA, OSH, SAV, and ASA. Although TCH does not perform the best in the remaining vegetation classifications, its overall performance is balanced and stable compared with other products, indicating that it effectively reduces the uncertainty of each product in different vegetation classifications. The balanced performance of TCH allows it to be used in most regions.

The spatial distribution of the combined product is consistent with that of the other three datasets, indicating that the product successfully captures the spatial difference of land ET. In addition, the combined product avoids the uncertain influence of the time-change trend of ERA5-Land, which also shows that it effectively captures the difference of the time-change trend of land-ET products. TCH also effectively reduces the uncertainty of the time-change trend of ET in spatial distribution. The equilibrium performance of ET products after fusion is helpful to accurately quantify hydrothermal balance, and it can provide basic data for the study of ET trends caused by global climate change.

**Supplementary Materials:** The following supporting information can be downloaded at: https://www.mdpi.com/article/10.3390/rs16010028/s1, Table S1: EC sites.

**Author Contributions:** Conceptualization, Z.C. and Y.Z.; methodology, Z.C. and Y.Z.; data curation, Z.C.; analysis work, Z.C.; writing—original draft preparation, Z.C.; writing—review and editing: Y.Z., A.W., J.W. and C.L.; supervision, Y.Z., A.W., J.W. and C.L.; project administration, Y.Z., A.W. and J.W.; funding acquisition, A.W. and J.W. All authors have read and agreed to the published version of the manuscript.

**Funding:** This research was funded by the National Key Research and Development Program of China (2022YFF1300501) and the National Natural Science Foundation of China (32271873, 32171873, and 31971728).

**Data Availability Statement:** A convenience copy of the fused global land evapotranspiration product available at https://zenodo.org/records/10065351, accessed on 2 November 2023.

**Conflicts of Interest:** The authors declare no conflicts of interest.

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
