# Peer review of "Uncertainty Analysis and Data Fusion of Multi-Source Land Evapotranspiration Products Based on the TCH Method"

_remotesensing, doi:10.3390/rs16010028_

Round 1

Reviewer 1 Report

Comments and Suggestions for Authors

General comments: This paper has fused three ET datasets by using the uncertainties-based TCP scheme and fluxes site observations for hopefully better ET production. The topic of this paper is impressive, and the overall written English is fine, however, some issues related to logic and technology need to be enhanced and clarified to sharpen the novelty of this work as follows.

Comment 1: The English language seems fine. The work has the potential to be published but before it should be considered for publication, it has to pass through professional proofreading, and all the highlighted points below need to be corrected and implemented.

Comment 2: I suggest the authors revise the introduction of the study per the comments raised. The authors can also use the following points below as a guideline to help them come out with an exciting introduction that is more scientific.

Background & Significance: (What general background does the reader need to understand the manuscript and how important is it in the context of scientific research).

Problem definition: (What are the research questions to fill in the gaps of the existing knowledge body or methodology )?

Motivations & Objectives: (Why are you conducting the study and what are the goals to achieve?)

Comment 3. For Section 2:

For data processing in section 2.1, the ERA5 upscale is fine, but the MERRA downscale seems too rough and unconvinced because this unmatched scale can directly affect your results. Thus, can the authors clearly explain their internal relations during your result and method sections?

For section 2.2.1, firstly, analyze the uncertainty of the monthly scale, and then determine the uncertainty of vegetation classification. Is that the case?

For section 2.2, the TCP method only uses grid products but not site observations, why? Whats the point of the verification with site observation if they are not actually included?  

Comment 4. For Section 3:

In Lines 252-253, in fact, ERA5 has less data (than the other two) over Tibet, does this is the direct reason for its high uncertainties over this specific region?

In Lines 442-450, according to your verification, the TCP method seems to rely more on the quality of the initial data itself, thus, does the fused datasets have been improved when compared to the initial data, especially for broader applications?   

Comment 5: For Section 4, the conclusion is too short and needs to be supplemented. Is the fused datasets in this article available and what are the application prospects? Especially compared to ERA5 and GLDAS data, the application in those areas will be better.

Comment 6: I strongly suggest separate sections for discussion, which should include a dialectical summary of the explanations underlying these characteristics during their findings.  

Comment 7: Also, several writing problems should be noticed below.

1-It is recommended not to use abbreviations in the abstract

2-Incorrect citation format in many places in this paper.

3-There seems to be a problem with the dashes in the formulas (7)~(11).

4-Line 253, broken links.

5-Figure 7, wrong greater-than in the legend.

Comments on the Quality of English Language

The English language seems fine. The work has the potential to be published but before it should be considered for publication, it has to pass through professional proofreading, and all the comments need to be corrected and implemented.

Reviewer 2 Report

Comments and Suggestions for Authors

In this study the authors create a new ET product out of 3 commonly available datasets to possibly improve its estimate for modeling purposes and evaluate the incorporated data to determine its uncertainty. The new dataset is at a large spatial resolution and therefore limits its utility to large area modeling. The introduction frames the purpose of the study well and provides inside on the advantages and disadvantages. The methods section adequately describes the methods used to create and evaluate the newly created ET dataset. The results and discussion section are well presented but could benefit from some clarification on sections 3.2, 3.3, and 3.4. which are addressed in the attached pdf. The graphics are very informative. Different color choices to make the results better readable or even more inclusive would improve the quality of the figures. Specific comments are in the attached pdf. The conclusions are very informative but lack some more explanation on what we could learn from it, why does the science community need this new product and what can it be used for. What did you use it for in your research? Text edits and specific comments are found in the file to improve the manuscript’s quality.

Comments on the Quality of English Language

The manuscript reads well with some minor edits required. The main issue was the references not being correctly presented in the manuscript.

Reviewer 3 Report

Comments and Suggestions for Authors

The manuscript is well written, there are some little problems need to be solved.

Need to check the reference format in the main text.

The use of RMSE and RMSD needs to be unified.

TCH is used in this study for the fusion of three global ET products, add some discussions for the fusion of more than three ET products.

Fig 2: Will the data product of MERRA-2 have any impact on the statistical results in Fig 2d as it has no value in Greenland. Also for the Caspian Sea.

As you mentioned "The uncertainty caused by the scaling effect is caused by the mismatch between the spatial resolution of the ET product and the flux tower." So can you consider comparing ET products with FLUXCOM products?

Round 2

Reviewer 2 Report

Comments and Suggestions for Authors

Thank you for the updated manuscript.